# Comparison of Six Measures of Genetic Similarity of Interspecific *Brassicaceae* Hybrids F_2_ Generation and Their Parental Forms Estimated on the Basis of ISSR Markers

**DOI:** 10.3390/genes15091114

**Published:** 2024-08-23

**Authors:** Jan Bocianowski, Janetta Niemann, Anna Jagieniak, Justyna Szwarc

**Affiliations:** 1Department of Mathematical and Statistical Methods, Poznań University of Life Sciences, Wojska Polskiego 28, 60-637 Poznań, Poland; 2Department of Genetics and Plant Breeding, Poznań University of Life Sciences, Dojazd 11, 60-632 Poznań, Poland; janetta.niemann@up.poznan.pl (J.N.); anna.jagieniak@up.poznan.pl (A.J.); justyna.szwarc@up.poznan.pl (J.S.)

**Keywords:** similarity measures, rapeseed, molecular markers, correlation, dendrogram

## Abstract

Genetic similarity determines the extent to which two genotypes share common genetic material. It can be measured in various ways, such as by comparing DNA sequences, proteins, or other genetic markers. The significance of genetic similarity is multifaceted and encompasses various fields, including evolutionary biology, medicine, forensic science, animal and plant breeding, and anthropology. Genetic similarity is an important concept with wide application across different scientific disciplines. The research material included 21 rapeseed genotypes (ten interspecific *Brassicaceae* hybrids of F_2_ generation and 11 of their parental forms) and 146 alleles obtained using 21 ISSR molecular markers. In the presented study, six measures for calculating genetic similarity were compared: Euclidean, Jaccard, Kulczyński, Sokal and Michener, Nei, and Rogers. Genetic similarity values were estimated between all pairs of examined genotypes using the six measures proposed above. For each genetic similarity measure, the average, minimum, maximum values, and coefficient of variation were calculated. Correlation coefficients between the genetic similarity values obtained from each measure were determined. The obtained genetic similarity coefficients were used for the hierarchical clustering of objects using the unweighted pair group method with an arithmetic mean. A multiple regression model was written for each method, where the independent variables were the remaining methods. For each model, the coefficient of multiple determination was calculated. Genetic similarity values ranged from 0.486 to 0.993 (for the Euclidean method), from 0.157 to 0.986 (for the Jaccard method), from 0.275 to 0.993 (for the Kulczyński method), from 0.272 to 0.993 (for the Nei method), from 0.801 to 1.000 (for the Rogers method) and from 0.486 to 0.993 (for the Sokal and Michener method). The results indicate that the research material was divided into two identical groups using any of the proposed methods despite differences in the values of genetic similarity coefficients. Two of the presented measures of genetic similarity (the Sokal and Michener method and the Euclidean method) were the same.

## 1. Introduction

Genetic diversity is the presence of different forms of genes found in all ecosystems [1,2]. This means that regardless of the type of environment in which specific species live, their population can exhibit variable genetic traits [3]. The essence of this diversity is that most of the variable gene forms are inherited from generation to generation [4]. The impact of genetic diversity directly affects the entire population [5,6,7]. It is crucial for the process of selection—a process in which some genotype variants have an advantage over others. Selection is an important mechanism that acts on animal and plant populations [8,9,10,11]. This effect can be observed both at the population level and at the individual level. The main factor driving selection is the availability of resources in a given environment [12,13]. Naturally, if only limited environmental resources are available, the only way to survive and thrive is to gain an advantage through a genotype that has greater suitability to the environment [14,15,16]. In the case of plants and animals, genetic diversity has a crucial impact on selection [17]. In the process of natural selection, the force of choice operates, where unfavorable traits are eliminated, and an individual with the most advantageous traits for a given environment has a greater chance of survival [18,19]. This natural mechanism favors variants that are best adapted to living in a particular environment [20,21].

As a result of genetic diversity, the selection process begins with the recurrent genetic components present in the population [22,23]. Over time, these become important factors driving the diverse selection process [24]. Existing adaptive capabilities, including genetic traits, determine a population’s ability to survive in challenging conditions and provide an advantage based on genetic components [25,26]. Consequently, a diverse genotype has a greater chance of survival and the ability to cope with environmental conditions through reproduction, leading to the emergence of new generations from this genotype [25]. Another factor influencing genotype selection is polymorphism, the phenomenon of genetic discontinuity manifested by the presence of different forms within a single species that differ functionally or structurally [27,28,29]. Polymorphism involves the formation of hierarchies and the division of functions within a population. It can occur at the phenotypic level or the genetic or molecular levels [30,31]. When the frequency of two (or more) variations in a given population is too high to be caused by mutation, the population is considered polymorphic. In general, greater genetic diversity in a population may result in a higher chance of gaining an advantage in genotype selection [30,32]. In reality, genetic diversity arising from the combination of environmental conditions and the influence of biological factors is crucial for the adaptation of a population to new conditions [33,34]. A high level of genetic diversity will help minimize the effects of changing environmental conditions, consequently increasing the chances of survival for an individual or population [2,35].

The results of genetic similarity (genetic variability) have great significance in making selection decisions, which is related to the choice of the method for calculating this characteristic [2]. The method of genetic similarity is used in genetics and zoology to determine the degree of relatedness between individuals [36,37,38]. It involves comparing the levels of concordance in the genome of a given species or gene or DNA sequences to determine the phylogenetic distance between individuals. This method is commonly used in studies of human genealogy, species phylogeny, and research on evolutionary mechanisms and the tracing of species origins [39,40]. The genetic similarity method contributes to evolutionary studies on the origins of species, their lineage, and migrations [41,42]. Along with other methods, it is used to assess the impact of environmental changes and biological processes on biodiversity. Due to the potentially multi-level analysis of all DNA sequences, this method is particularly useful in zoology and molecular biology.

Different measures of calculating genetic similarity represent a specific type of statistic that compares genetic data between individuals [43,44,45]. Each measure has its advantages and disadvantages and reflects a particular methodology. The commonly encountered measures of genetic similarity can be divided into four main categories: classical, contemporary, dendrographic, and index-based. Classical measures include standard units, mean components, and the inbreeding index [46]. The first is a measure of identity, while the other two reflect the genetic composition relative to a given subset or population. Contemporary measures of genetic similarity, such as jump length and interdependence level, reflect the relationship between individual data and general data [47]. These measures identify the tendencies to occur in groups and calculate the likelihood of specific objects appearing within a population. These measures are often used, particularly in studies of genetic authenticity and microorganism identity. Dendrographic measures are very similar to classical measures but reflect similarities and differences at the genetic level [48,49]. Dendrographic measures help in understanding the genetic relationships between populations and other groups or individuals [50]. Index-based measures are multidisciplinary measures that examine the data structure at the population level [51]. They utilize elements of previous measures, estimating levels of similarities and differences within a population. Among other things, they estimate the ratio of individuals sharing a common ancestor or provide a detailed view of kinship and mismatch levels within a population. The aim of this study was to compare six statistical measures (Euclidean, Jaccard, Kulczyński, Sokal and Michener, Nei, and Rogers) to assess the genetic similarities of 21 rapeseed objects (10 interspecific *Brassicaceae* hybrids of the F_2_ generation, and 11 of their parental forms) based on the observations of 146 alleles obtained using 21 ISSR molecular markers.

## 2. Materials and Methods

### 2.1. Plant Material

The plant material for the genetic similarity analysis consisted of twenty-one *Brassicaceae* genotypes, including ten various *Brassicaceae* F_2_ hybrids and eleven parental genotypes (Table 1). Interspecific hybrids of the F_2_ generation were developed at the Department of Genetics and Plant Breeding (Poznań University of Life Sciences) with the use of in vitro embryo cultures.

### 2.2. Molecular Analysis

Genomic DNA was extracted from young seedling leaves of the studied genotypes using the Genomic Mini AX Plant kit (A&ABiotechnology, Gdańsk, Poland) according to the manufacturer’s protocol. The DNA concentration and purity were determined using a DeNovix^®^ DS-11 Spectrophotometer (Tokyo, Japan). Prior to the PCR analysis, DNA samples were diluted to 50 ng μL^−1^ using Tris buffer. PCR reactions were performed in a total volume of 13.05 µL (6.25 µL of DreamTaq PCR Master Mix, 0.25 µL of primer, 5.55 µL of H_2_O, and 1 µL of DNA template) under the conditions indicated in Table 2. The primer sequences and annealing temperatures are presented in Table 3.

Separation of the amplification product was performed on 1.5% agarose gel stained with Midori Green Advance (6 µL per 100 mL) in 1× TBE buffer at 104 V. The gels were visualized using the BIO-RAD Molecular Imager^®^ (Hercules, CA, USA) Gel Doc^TM^ XR+ with Image Lab^TM^ v. 5.1 Software.

### 2.3. Measures of Genetic Similarity

The following six measures, most commonly used in practical research, were used to estimate the genetic similarity (GS) of the genotypes studied.

Euclidean measure [55]:(1)$$S_{E,AB}=1-\sqrt{\sum_{i=1}^{N}{\left(\frac{m_{iA}-m_{iB}}{N}\right)}^{2}},$$
where *N* represents the number of markers and $m_{iA}$ ($m_{iB}$) denotes the observation of the *i*-th allele of genotype *A* (genotype *B*).

The Jaccarda measure [56]:(2)$$S_{J,AB}=\frac{N_{AB}}{N_{A}+N_{B}-N_{AB}},$$
where $N_{A}$ denotes the number of alleles present in genotype *A*, $N_{B}$ denotes the number of alleles present in genotype *B*, and $N_{AB}$ denotes the number of alleles present in both genotype *A* and genotype *B*.

The Kulczyński measure [57]:(3)$$S_{K,AB}=\frac{N_{AB}\left(N_{A}+N_{B}\right)}{2N_{A}\cdot N_{B}}.$$

The Nei measure [58]:(4)$$S_{N,AB}=\frac{2N_{AB}}{N_{A}+N_{B}}.$$

The Rogers measure [59]:(5)$$S_{R,AB}=1-\frac{\left|N_{0B}-N_{0B}\right|}{N},$$
where $N_{A0}$ denotes the number of alleles present in genotype *A* and is simultaneously absent in genotype *B* and $N_{0B}$ denotes the number of alleles present in genotype *B* and is simultaneously absent in genotype *A*.

The Sokal and Michener measure [60]:(6)$$S_{SM,AB}=\frac{N_{AB}+N_{00}}{N},$$
where $N_{00}$ denotes the number of cases where the same alleles are simultaneously absent in genotypes *A* and *B*.

### 2.4. Statistical Analysis

Genetic similarity values were estimated between all pairs of examined genotypes using the six measures proposed above. For each genetic similarity measure, the average, minimum, maximum values, and coefficient of variation were calculated. Correlation coefficients between the genetic similarity values obtained from each measure were determined. The significance of correlation coefficients was tested at a level of *α* = 0.001. The obtained genetic similarity coefficients were used for the hierarchical clustering of objects using the unweighted pair group method with the arithmetic mean (UPGMA). The results of the clustering analyses were presented in the form of dendrograms for the respective measures. A multiple regression model was written for each method (Euclidean, Jaccard, Kulczynski, Sokal and Michener, Nei, and Rogers), where the independent variables were the remaining methods. For each model, the coefficient of multiple determination (R^2^) was calculated. All these analyses were conducted using the GenStat v. 23.1 statistical software package [61].

## 3. Results

The analyzed material was examined using 21 primers. The number of amplification products for individual primers ranged from four (for UBC889, ISSR856, ISSR857, and ISSR890) to 11 (for mstg13, UBC835, and UBC891). In total, 146 markers were obtained.

Table 4 presents the statistical characteristics of the genetic similarity coefficients obtained using six measures. The genetic similarity values estimated by the Euclidean method ranged from 0.486 (for two pairs of genotypes: SA-BN and SA-BJ) to 0.993 (for three pairs of genotypes: BN × BRc-BN × BJ, BN × SA-BN × BRc, and BN × BT-BN × SA) (Table 5), with an average of 0.709 (Table 4). The genetic similarity coefficients calculated using the Euclidean method exhibited moderate variability, with a coefficient of variation of 24.21% (Figure 1, Table 4).

The smallest genetic similarity coefficients were obtained for the measure proposed by Jaccard (with an average of 0.508). These coefficients exhibited the highest variability (54.15%) among all measures used (Table 4, Figure 1). The lowest genetic similarity, according to Jaccard, was observed between the pair BJ-SA (0.157), while the highest was between the pair BN × BJ-BN × BRc (0.986) (Table 5).

Slightly larger genetic similarity coefficients were obtained for the measures proposed by Kulczyński and Nei compared to Jaccard. The results obtained with these two methods were similar in terms of both the values of genetic similarity coefficients (with average values of 0.640 and 0.633, respectively) (Table 4) and their variability (the coefficients of variation were 34.46% and 35.43%, respectively) (Table 4, Figure 1). The genetic similarity coefficients by Kulczyński ranged from 0.275 (between BJ and SA) to 0.993 (between BN × BJ and BN × BRc) (Table 6). Meanwhile, genetic similarity, calculated using the Nei method, ranged from 0.272 (between BJ and SA) to 0.993 (between BN × BJ and BN × BRc) (Table 6).

Significantly different results were obtained when using the Rogers measure (Table 4 and Table 7, Figure 1). The genetic similarity coefficients exhibited the lowest variability: 6.14% (Table 4) and ranged from 0.801 (for two genotype pairs: BF-BN × BJ and BF-BN × BRp) to 1.000 (for 23 genotype pairs) (Table 7). The average value of the Rogers coefficients was 0.929 (Table 4). The minimum value of the genetic similarity coefficients, estimated by this measure, was higher than the average values obtained for the other measures.

The genetic similarity coefficients calculated using the Sokal and Michener measure were identical to the coefficients calculated using the Euclidean method (Table 5 and Table 7). The minimum value was twice the minimum value obtained using the Kulczyński or Nei measures and four times when the Jaccard measure was used for comparison.

Figure 2 depicts the linear correlation coefficients between the genetic similarity values obtained using six measures. The results indicate statistically significant correlations between all applied genetic similarity measures (at *α* = 0.001 level). It is noteworthy that the genetic similarity coefficients calculated by the Kulczyński and Nei methods show perfect correlation (correlation coefficient was 1.000). A similar result was obtained for the Euclidean vs. the Sokal and Michener measures (Figure 2). However, this is due to the identical values obtained from these two methods.

The obtained genetic similarity coefficients were used for the hierarchical clustering of objects using the average linkage method. The results of the clustering are presented in the form of dendrograms for the respective measures (Figure 3). The clustering of the genotypes based on the genetic similarity coefficients according to the Euclidean measure (Figure 3A) and Sokal and Michener measure (Figure 3F) was identical due to the identical similarity values obtained from these two methods. The clustering of the genotypes based on the genetic similarity coefficients according to the Kulczyński measure (Figure 3C) and Nei measure (Figure 3D) was nearly identical. A very similar clustering of objects to the aforementioned two measures was obtained based on Jaccard coefficients (Figure 3B). It is noteworthy that based on the genetic similarity coefficients estimated by all six measures, the examined rapeseed genotypes were clustered similarly into two clusters. One of these clusters comprised ten rapeseed hybrids, and the other comprised eleven parental forms.

## 4. Discussion

An important element in the accurate assessment of genetic similarity in plant material is the choice of molecular analysis technique, the number of molecular markers used in the study, and the appropriate statistical measure to estimate the relationships between studied genotypes [62]. Genetic variability among studied objects is defined based on the frequency of the DNA polymorphism obtained using different types of molecular markers. Various measures can be applied in estimating the genetic similarity/distance based on the results obtained from molecular markers. In studies of *Brassica* crops, the Nei and Li measure [63] has been most commonly used for Indian mustard (*B. juncea*) [64], cabbage (*B. oleracea*) [65], and rapeseed (*B. napus*) [66]. The Jaccard measure has been applied in studies on rapeseed [67] and cabbage [68,69], among others. To a lesser extent, in rapeseed studies, the Sokal and Michener measure has been utilized [70]. The Rogers measure has been used to assess the genetic variability in crops such as maize (*Zea mays* L.) [71,72] and sorghum (*Sorghum bicolor* L.) [73].

In the literature on the subject, little attention has been devoted to the theoretical comparisons of genetic similarity measures. Lamboy [74] compared the Nei and Jaccard measures, pointing out Nei’s measure as less biased for analyzing closely related organisms. Five of the genetic similarity measures used in the present study (excluding the Rogers measure) yielded very similar results. However, determining the best measure is not feasible. Nearly identical clustering of genotypes suggests that any of the proposed methods would be suitable for the overall characterization of the studied materials. Mohammadi and Prasanna [62] highlight Rogers’ genetic distance measure as useful for estimating genetic similarity based on the co-dominant molecular markers, where the amplification product can be equated with alleles. This is corroborated by Gauthier et al. [75] in their study of European maize populations, where both Rogers’ measure utilizing allele frequencies and Nei and Li’s measure based on a binary system (where 1 indicates allele presence and 0 is its absence) were applied. Similarly, Lombard et al. [76] demonstrated the utility of Rogers’ measure in assessing homogeneity within three rapeseed varieties, assuming the tested material was homozygous, and each marker was treated as a diallelic locus. Similar findings were reported by Lee et al. [77] and Dudley et al. [78] for homozygous maize breeding lines, as well as by Jordan et al. [73] for the sorghum lines analyzed using co-dominant markers.

The distinctly different similarity coefficient values obtained using the Rogers method stem from the fact that its calculation principle differs from the other proposed methods. Rogers’ method considers the number of alleles present only in one of the compared genotypes, whereas the other methods are based on the number of alleles present simultaneously in both compared genotypes. Previous studies on generated data [79] as well as real data [80] indicate a lack of significant correlation between Rogers’ genetic similarity measure and the other measures. The significant correlation obtained in this study may be a result of a large number of markers obtained, significantly exceeding the number of genotype observations considered in the aforementioned studies. Furthermore, it is noteworthy that each of the six applied measures yielded two identical clusters (hybrids and their parental forms) as a consequence of clustering using the UPGMA method. Rogers’ GS coefficients also distinguished these two groups despite the significantly different values compared to those obtained by the other methods. We performed clustering not only with the UPGMA method but also with other methods, such as maximum likelihood, neighbor-joining, minimum evolution, and maximum parsimony. The results we obtained were very similar, and some were identical.

Duarte et al. [81] obtained dendrograms for bean varieties with the same structure using UPGMA clustering based on Jaccard’s and Nei’s coefficients. Lombard et al. [70] applied the genetic similarity (GS) coefficients of Jaccard (J), Sokal and Michener (SM), and modified Sokal and Michener (MSM) to compare 83 spring and winter rapeseed varieties, whose diversity was assessed using AFLP. All of the tested similarity coefficients showed significant correlations (*r* = 0.96 for J and MSM, 0.97 for SM and MSM, and 0.98 for J and SM at the significance level of 0.001). The genetic similarity calculated using these three coefficients allowed for a very similar estimation of relationships among the studied varieties. A previous study [82] showed that the type of markers did not affect the evaluation of the genetic similarity coefficients of CMS *ogura* F_1_ hybrids of winter oilseed rape (*B. napus* L.) parental lines.

## 5. Conclusions

The purpose of this study was to compare six statistical measures (Euclidean, Jaccard, Kulczynski, Sokal and Michener, Nei, and Rogers) for assessing the genetic similarity of 21 rapeseed objects (10 interspecific hybrids of *Brassicaceae* of generation F_2_ and 11 of their parental forms) based on the observations of 146 alleles obtained using 21 ISSR molecular markers. All the mentioned measures define different categories of genetic similarity assessment and coexist with other techniques, such as gene sequencing and flora and fauna analyses. Choosing the appropriate type of measure requires a thorough understanding of its purpose and calculation technique to provide an assessment that is precise and comprehensive enough. Significantly different results were obtained when using the Rogers measure. The research material was divided into two identical groups using any of the methods despite differences in the genetic similarity coefficient values. Two of the presented genetic similarity measures (Sokal and Michener as well as Euclidean) were identical.

## Figures and Tables

**Figure 1 genes-15-01114-f001:**
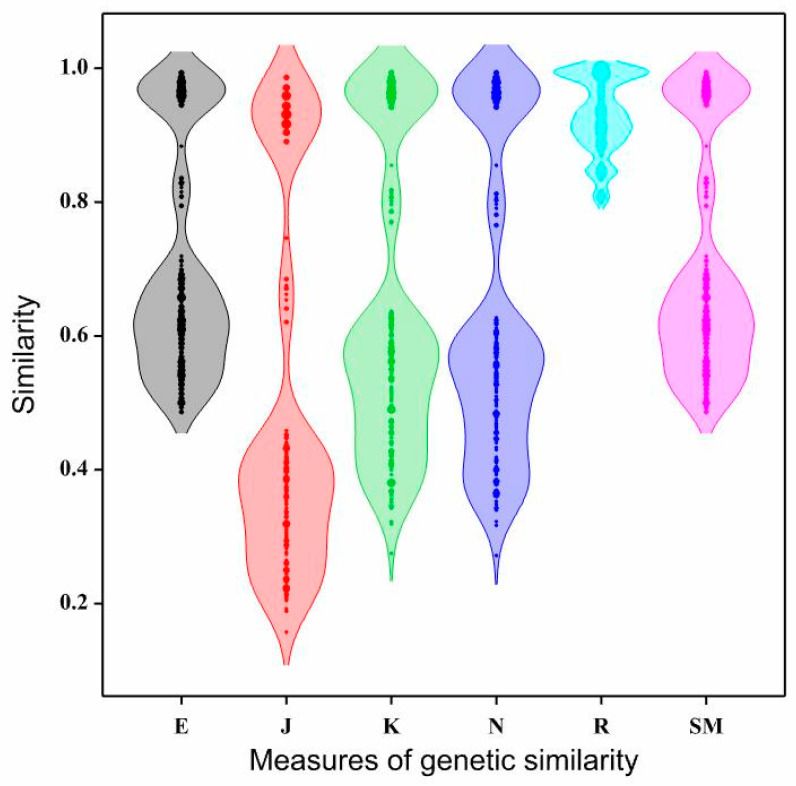
Density plots of coefficients of genetic similarity of interspecific *Brassicaceae* hybrids and their parental forms using six measures of genetic similarity: E—Euclidean, J—Jaccard, K—Kulczyński, N—Nei, R—Rogers, SM—Sokal and Michener.

**Figure 2 genes-15-01114-f002:**
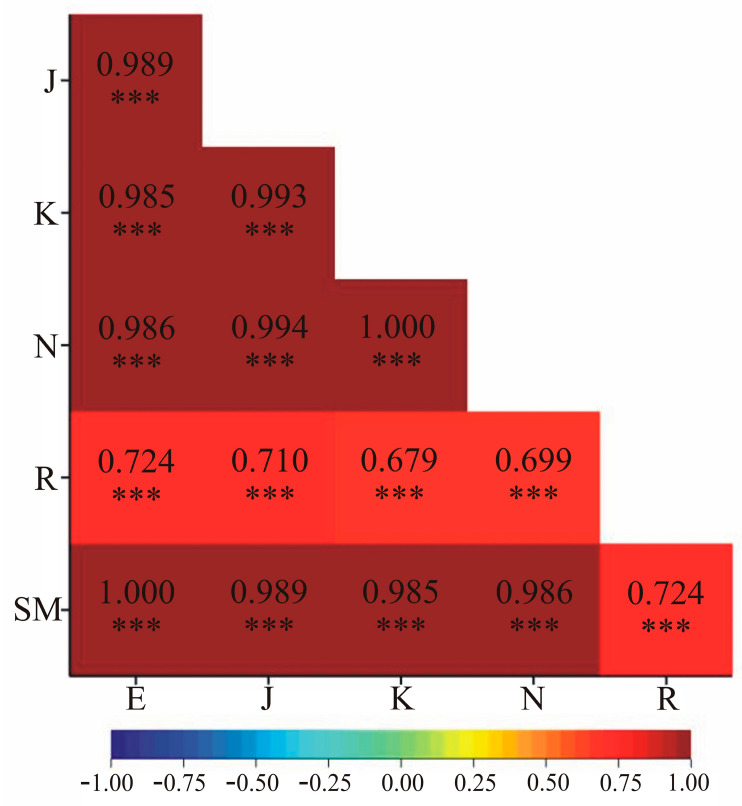
Linear correlation coefficients between pairs of measures of genetic similarity of the studied interspecific *Brassicaceae* hybrids and their parental forms: E—Euclidean, J—Jaccard, K—Kulczyński, N—Nei, R—Rogers, SM—Sokal and Michener. The coefficient of multiple determination (R^2^) for each method were equal: 1.00 (for Euclidean), 0.99 (for Jaccard), 1.00 (for Kulczyński), 1.00 (for Nei), 0.959 (for Rogers), and 1.00 (for Sokal and Michener). *** *p* < 0.001.

**Figure 3 genes-15-01114-f003:**
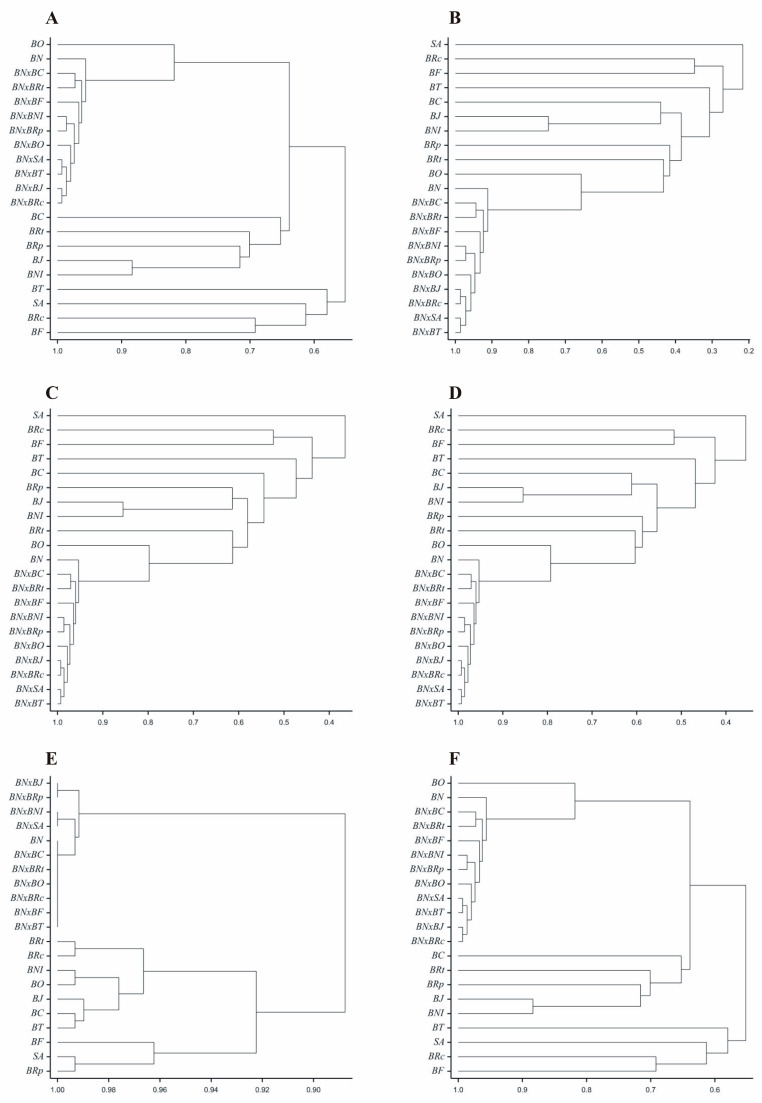
Dendrograms of genetic similarity of 21 *Brassicaceae* genotypes constructed using six measures based on 146 ISSR markers. (**A**)—Euclidean, (**B**)—Jaccard, (**C**)—Kulczyński, (**D**)—Nei, (**E**)—Rogers, (**F**)—Sokal and Michener.

**Table 1 genes-15-01114-t001:** Plant material used in this study for the analysis of genetic similarity based on ISSR markers.

Maternal Form	Paternal Forms	Hybrids of F_2_ Generation
*Brassica napus* (BN)	*Brassica carinata* (BC)	*B. napus* × *B. carinata* (BN × BC)
	*Brassica rapa* ssp. *trilocularis* (BRt)	*B. napus* × *B. rapa* ssp. *trilocularis* (BN × BRt)
	*Brassica rapa* ssp. *chinensis* (BRc)	*B. napusx* × *B. rapa* ssp. *chinensis* (BN × BRc)
	*Brassica fruticulosa* (BF)	*B. napus* × *B. fruticulosa* (BN × BF)
	*Brassica rapa* ssp. *pekinensis* (BRp)	*B. napus* × *B. rapa* ssp. *pekinensis* (BN × BRp)
	*Brassica nigra* (BNI)	*B. napus* × *B. nigra* (BN × BNI)
	*Brassica juncea* (BJ)	*B. napus* × *B. juncea* (BN × BJ)
	*Sinapis alba* (SA)	*B. napus* × *S. alba* (BN × SA)
	*Brassica tournefortii* (BT)	*B. napus* × *B. rapa* ssp. *tournefortii* (BN × BT)
	*Brassica oleracea* var. *alboglabra* (BO)	*B. napus* × *B. oleracea* var. *alboglabra* (BN × BO)

**Table 2 genes-15-01114-t002:** PCR amplification profiles for tested primers.

Reaction Profile	Primers	Parameters	Temperature	Number of Cycles	Source
1	mstg 1, mstg 4, mstg 12, mstg 13, mstg 38	Initial denaturation	94 °C 30 s	×1	[52]
		Denaturation	94 °C 30 s	×40	
		Primers annealing	60 °C * 30 s		
		Elongation	72 °C 1 min		
		Final elongation	72 °C 5 min	×1	
2	UBC812, UBC840, UBC845	Initial denaturation	95 °C 2 min	×1	[53]
		Denaturation	93 °C 20 s	×40	
		Primers annealing	52 °C * 1 min		
		Elongation	72 °C 20 s		
		Final elongation	72 °C 6 min	×1	
3	UBC891, ISSR-840, ISSR-856, ISSR-857, ISSR-887, ISSR-888, ISSR-889, ISSR-890, scm51, P1, P2, P4, P5	Initial denaturation	94 °C 5 min	×1	[54]
		Denaturation	94 °C 45 s	×35	
		Primers annealing	50 °C * 45 s		
		Elongation	72 °C 90 s		
		Final elongation	72 °C 7 min	×1	

* Primer annealing temperature was modified for each PCR reaction (Table 3).

**Table 3 genes-15-01114-t003:** List of ISSR primers used in this study.

ISSR Primer	Sequence 5′-3′	Annealing Temperature [°C]
mstg 1	(TA)_3_(TG)_9_	53.9
mstg 4	(TA)_3_(TG)_10_	60
mstg 12	(TG)_7_	54
mstg 13	(TG)_7_CG(TG)_2_	59.7
mstg 38	(TG)_3_TT(TG)_5_	53.9
UBC835	(AG)_8_YC	51.5
UBC843	(CT)_8_RA	48.1
UBC889	DBD(AC)_7_	51.5
UBC891	HVH(TG)_7_	51.5
scm 51	(AGG)_5_	52.8
P1	GAG(CAA)_5_	49.7
P2	CTG(GT)_8_	56.5
P4	(AG)_8_GTG	52.8
P5	(GA)_8_ACC	51.5
ISSR-840	(GA)_8_YT	48.4
ISSR-856	(GA)_8_YA	48.4
ISSR-857	(GA)_8_YG	55.3
ISSR-887	DVD(TC)_7_	47.8
ISSR-888	BDB(CA)_7_	48.4
ISSR-889	DBD(AC)_7_	48.1
ISSR-890	VHV(GT)_7_	48.4

**Table 4 genes-15-01114-t004:** Statistical characteristics of genetic similarity of interspecific *Brassicaceae* hybrids F_2_ generation and their parental forms.

Genetic Similarity Measure	Minimal Value	Mean Value	Maximum Value	Coefficient of Variability [in %]
Euclidean	0.486	0.709	0.993	24.21
Jaccard	0.157	0.508	0.986	54.15
Kulczyński	0.275	0.640	0.993	34.46
Nei	0.272	0.633	0.993	35.43
Rogers	0.801	0.929	1.000	6.14
Sokal and Michener	0.486	0.709	0.993	24.21

**Table 5 genes-15-01114-t005:** Genetic similarity coefficients between the examined genotypes calculated using the Euclidean measure (above diagonal) and Jaccard measure (below diagonal).

Genotype	BN	BC	BRt	BJ	BNI	BO	BRc	SA	BRp	BF	BT	BN × BC	BN × BRt	BN × BJ	BN × BNI	BN × BO	BN × BRc	BN × SA	BN × BRp	BN × BF	BN × BT
BN	**1.000**	0.596	0.658	0.603	0.610	0.808	0.541	0.486	0.644	0.534	0.562	0.959	0.959	0.952	0.952	0.945	0.959	0.952	0.966	0.959	0.959
BC	0.359	**1.000**	0.582	0.692	0.699	0.610	0.658	0.548	0.637	0.637	0.500	0.637	0.637	0.603	0.603	0.596	0.596	0.603	0.616	0.610	0.610
BRt	0.419	0.282	**1.000**	0.685	0.706	0.644	0.555	0.555	0.712	0.589	0.521	0.671	0.671	0.678	0.678	0.671	0.685	0.692	0.664	0.658	0.685
BJ	0.370	0.430	0.410	**1.000**	0.884	0.616	0.623	0.486	0.712	0.630	0.589	0.616	0.616	0.637	0.623	0.644	0.630	0.623	0.623	0.616	0.630
BNI	0.387	0.450	0.449	0.746	**1.000**	0.610	0.658	0.521	0.719	0.623	0.582	0.610	0.623	0.630	0.603	0.610	0.623	0.616	0.616	0.610	0.623
BO	0.641	0.337	0.366	0.349	0.352	**1.000**	0.527	0.541	0.699	0.589	0.589	0.795	0.795	0.829	0.829	0.836	0.836	0.829	0.815	0.808	0.822
BRc	0.287	0.367	0.235	0.329	0.383	0.233	**1.000**	0.575	0.596	0.692	0.555	0.555	0.541	0.507	0.534	0.514	0.514	0.521	0.548	0.541	0.527
SA	0.211	0.214	0.207	0.157	0.205	0.221	0.225	**1.000**	0.569	0.651	0.527	0.500	0.500	0.493	0.507	0.500	0.500	0.507	0.493	0.514	0.500
BRp	0.381	0.321	0.409	0.425	0.446	0.413	0.253	0.192	**1.000**	0.671	0.658	0.658	0.671	0.678	0.678	0.685	0.685	0.692	0.664	0.658	0.685
BF	0.236	0.293	0.221	0.290	0.295	0.250	0.348	0.261	0.294	**1.000**	0.658	0.562	0.548	0.527	0.541	0.534	0.534	0.541	0.541	0.548	0.548
BT	0.319	0.207	0.214	0.302	0.307	0.310	0.244	0.188	0.342	0.315	**1.000**	0.548	0.562	0.555	0.582	0.575	0.562	0.569	0.569	0.562	0.562
BN × BC	0.917	0.405	0.435	0.385	0.387	0.620	0.301	0.223	0.398	0.264	0.305	**1.000**	0.973	0.952	0.966	0.945	0.959	0.966	0.980	0.959	0.973
BN × BRt	0.917	0.405	0.435	0.385	0.402	0.620	0.287	0.223	0.415	0.250	0.319	0.944	**1.000**	0.952	0.966	0.945	0.959	0.966	0.980	0.959	0.973
BN × BJ	0.904	0.370	0.447	0.411	0.413	0.675	0.258	0.221	0.427	0.233	0.316	0.904	0.904	**1.000**	0.973	0.980	0.993	0.986	0.959	0.966	0.980
BN × BNI	0.903	0.363	0.441	0.389	0.376	0.671	0.277	0.226	0.420	0.239	0.337	0.930	0.930	0.944	**1.000**	0.980	0.980	0.986	0.986	0.966	0.980
BN × BO	0.890	0.359	0.435	0.416	0.387	0.684	0.260	0.223	0.432	0.236	0.333	0.890	0.890	0.958	0.957	**1.000**	0.986	0.980	0.966	0.959	0.973
BN × BRc	0.917	0.359	0.452	0.400	0.402	0.684	0.260	0.223	0.432	0.236	0.319	0.917	0.917	0.986	0.957	0.971	**1.000**	0.993	0.966	0.973	0.986
BN × SA	0.903	0.363	0.458	0.389	0.391	0.671	0.263	0.226	0.438	0.239	0.323	0.930	0.930	0.971	0.971	0.957	0.986	**1.000**	0.973	0.966	0.993
BN × BRp	0.931	0.385	0.430	0.396	0.398	0.654	0.298	0.221	0.410	0.247	0.330	0.958	0.958	0.918	0.971	0.931	0.931	0.944	**1.000**	0.966	0.980
BN × BF	0.917	0.374	0.419	0.385	0.387	0.641	0.287	0.237	0.398	0.250	0.319	0.917	0.917	0.931	0.930	0.917	0.944	0.930	0.931	**1.000**	0.973
BN × BT	0.917	0.374	0.452	0.400	0.402	0.662	0.274	0.223	0.432	0.250	0.319	0.944	0.944	0.958	0.957	0.944	0.971	0.986	0.958	0.944	**1.000**

**Table 6 genes-15-01114-t006:** Genetic similarity coefficients between the examined genotypes were calculated using the Kulczyński measure (above diagonal) and Nei measure (below diagonal).

Genotype	BN	BC	BRt	BJ	BNI	BO	BRc	SA	BRp	BF	BT	BN × BC	BN × BRt	BN × BJ	BN × BNI	BN × BO	BN × BRc	BN × SA	BN × BRp	BN × BF	BN × BT
BN	**1.000**	0.534	0.601	0.545	0.561	0.786	0.455	0.362	0.572	0.408	0.490	0.957	0.957	0.950	0.949	0.942	0.957	0.949	0.964	0.957	0.957
BC	0.528	**1.000**	0.441	0.602	0.621	0.505	0.538	0.356	0.489	0.465	0.342	0.582	0.582	0.546	0.537	0.534	0.534	0.537	0.563	0.550	0.550
BRt	0.590	0.440	**1.000**	0.583	0.622	0.537	0.381	0.345	0.582	0.368	0.352	0.617	0.617	0.630	0.621	0.617	0.634	0.638	0.613	0.601	0.634
BJ	0.540	0.602	0.582	**1.000**	0.855	0.517	0.497	0.275	0.602	0.461	0.464	0.561	0.561	0.589	0.564	0.593	0.577	0.564	0.573	0.561	0.577
BNI	0.558	0.621	0.620	0.855	**1.000**	0.521	0.556	0.346	0.626	0.472	0.471	0.561	0.576	0.588	0.549	0.561	0.576	0.565	0.573	0.561	0.576
BO	0.781	0.504	0.536	0.517	0.521	**1.000**	0.380	0.368	0.593	0.413	0.474	0.770	0.770	0.812	0.807	0.818	0.818	0.807	0.797	0.786	0.802
BRc	0.446	0.537	0.381	0.495	0.554	0.378	**1.000**	0.369	0.405	0.524	0.393	0.472	0.455	0.419	0.441	0.422	0.422	0.424	0.469	0.455	0.438
SA	0.348	0.353	0.343	0.272	0.340	0.362	0.367	**1.000**	0.323	0.415	0.319	0.380	0.380	0.378	0.383	0.380	0.380	0.383	0.378	0.399	0.380
BRp	0.552	0.485	0.580	0.596	0.617	0.585	0.404	0.323	**1.000**	0.457	0.513	0.590	0.608	0.622	0.612	0.626	0.626	0.630	0.605	0.590	0.626
BF	0.382	0.454	0.362	0.449	0.455	0.400	0.516	0.414	0.455	**1.000**	0.490	0.447	0.428	0.406	0.411	0.408	0.408	0.411	0.425	0.428	0.428
BT	0.484	0.342	0.352	0.464	0.470	0.474	0.393	0.317	0.510	0.479	**1.000**	0.474	0.490	0.487	0.510	0.507	0.490	0.493	0.503	0.490	0.490
BN × BC	0.957	0.576	0.607	0.556	0.558	0.766	0.463	0.365	0.569	0.418	0.468	**1.000**	0.971	0.950	0.964	0.942	0.957	0.964	0.979	0.957	0.971
BN × BRt	0.957	0.576	0.607	0.556	0.574	0.766	0.446	0.365	0.586	0.400	0.484	0.971	**1.000**	0.950	0.964	0.942	0.957	0.964	0.979	0.957	0.971
BN × BJ	0.950	0.540	0.618	0.583	0.585	0.806	0.410	0.362	0.598	0.378	0.480	0.950	0.950	**1.000**	0.971	0.979	0.993	0.986	0.957	0.964	0.979
BN × BNI	0.949	0.532	0.612	0.560	0.547	0.803	0.433	0.368	0.591	0.385	0.504	0.964	0.964	0.971	**1.000**	0.978	0.978	0.985	0.986	0.964	0.978
BN × BO	0.942	0.528	0.607	0.587	0.558	0.813	0.413	0.365	0.603	0.382	0.500	0.942	0.942	0.978	0.978	**1.000**	0.986	0.978	0.964	0.957	0.971
BN × BRc	0.957	0.528	0.623	0.571	0.574	0.813	0.413	0.365	0.603	0.382	0.484	0.957	0.957	0.993	0.978	0.986	**1.000**	0.993	0.964	0.971	0.986
BN × SA	0.949	0.532	0.628	0.560	0.563	0.803	0.417	0.368	0.609	0.385	0.488	0.964	0.964	0.986	0.985	0.978	0.993	**1.000**	0.971	0.964	0.993
BN × BRp	0.964	0.556	0.602	0.567	0.569	0.791	0.459	0.362	0.581	0.396	0.496	0.978	0.978	0.957	0.986	0.964	0.964	0.971	**1.000**	0.964	0.979
BN × BF	0.957	0.544	0.590	0.556	0.558	0.781	0.446	0.383	0.569	0.400	0.484	0.957	0.957	0.964	0.964	0.957	0.971	0.964	0.964	**1.000**	0.971
BN × BT	0.957	0.544	0.623	0.571	0.574	0.797	0.430	0.365	0.603	0.400	0.484	0.971	0.971	0.978	0.978	0.971	0.986	0.993	0.978	0.971	**1.000**

**Table 7 genes-15-01114-t007:** Genetic similarity coefficients between the examined genotypes were calculated using the Rogers measure (above diagonal) and the Sokal and Michener measure (below diagonal).

Genotype	BN	BC	BRt	BJ	BNI	BO	BRc	SA	BRp	BF	BT	BN × BC	BN × BRt	BN × BJ	BN × BNI	BN × BO	BN × BRc	BN × SA	BN × BRp	BN × BF	BN × BT
BN	**1.000**	0.911	0.890	0.918	0.938	0.932	0.884	0.843	0.849	0.808	0.904	1.000	1.000	0.993	0.993	1.000	1.000	0.993	0.993	1.000	1.000
BC	0.596	**1.000**	0.980	0.993	0.973	0.980	0.973	0.932	0.938	0.897	0.993	0.911	0.911	0.904	0.918	0.911	0.911	0.918	0.904	0.911	0.911
BRt	0.658	0.582	**1.000**	0.973	0.952	0.959	0.993	0.952	0.959	0.918	0.986	0.890	0.890	0.884	0.897	0.890	0.890	0.897	0.884	0.890	0.890
BJ	0.603	0.692	0.685	**1.000**	0.980	0.986	0.966	0.925	0.932	0.890	0.986	0.918	0.918	0.911	0.925	0.918	0.918	0.925	0.911	0.918	0.918
BNI	0.610	0.699	0.706	0.884	**1.000**	0.993	0.945	0.904	0.911	0.870	0.966	0.938	0.938	0.932	0.945	0.938	0.938	0.945	0.932	0.938	0.938
BO	0.808	0.610	0.644	0.616	0.610	**1.000**	0.952	0.911	0.918	0.877	0.973	0.932	0.932	0.925	0.938	0.932	0.932	0.938	0.925	0.932	0.932
BRc	0.541	0.658	0.555	0.623	0.658	0.527	**1.000**	0.959	0.966	0.925	0.980	0.884	0.884	0.877	0.890	0.884	0.884	0.890	0.877	0.884	0.884
SA	0.486	0.548	0.555	0.486	0.521	0.541	0.575	**1.000**	0.993	0.966	0.938	0.843	0.843	0.836	0.849	0.843	0.843	0.849	0.836	0.843	0.843
BRp	0.644	0.637	0.712	0.712	0.719	0.699	0.596	0.569	**1.000**	0.959	0.945	0.849	0.849	0.843	0.856	0.849	0.849	0.856	0.843	0.849	0.849
BF	0.534	0.637	0.589	0.630	0.623	0.589	0.692	0.651	0.671	**1.000**	0.904	0.808	0.808	0.801	0.815	0.808	0.808	0.815	0.801	0.808	0.808
BT	0.562	0.500	0.521	0.589	0.582	0.589	0.555	0.527	0.658	0.658	**1.000**	0.904	0.904	0.897	0.911	0.904	0.904	0.911	0.897	0.904	0.904
BN × BC	0.959	0.637	0.671	0.616	0.610	0.795	0.555	0.500	0.658	0.562	0.548	**1.000**	1.000	0.993	0.993	1.000	1.000	0.993	0.993	1.000	1.000
BN × BRt	0.959	0.637	0.671	0.616	0.623	0.795	0.541	0.500	0.671	0.548	0.562	0.973	**1.000**	0.993	0.993	1.000	1.000	0.993	0.993	1.000	1.000
BN × BJ	0.952	0.603	0.678	0.637	0.630	0.829	0.507	0.493	0.678	0.527	0.555	0.952	0.952	**1.000**	0.986	0.993	0.993	0.986	1.000	0.993	0.993
BN × BNI	0.952	0.603	0.678	0.623	0.603	0.829	0.534	0.507	0.678	0.541	0.582	0.966	0.966	0.973	**1.000**	0.993	0.993	1.000	0.986	0.993	0.993
BN × BO	0.945	0.596	0.671	0.644	0.610	0.836	0.514	0.500	0.685	0.534	0.575	0.945	0.945	0.980	0.980	**1.000**	1.000	0.993	0.993	1.000	1.000
BN × BRc	0.959	0.596	0.685	0.630	0.623	0.836	0.514	0.500	0.685	0.534	0.562	0.959	0.959	0.993	0.980	0.986	**1.000**	0.993	0.993	1.000	1.000
BN × SA	0.952	0.603	0.692	0.623	0.616	0.829	0.521	0.507	0.692	0.541	0.569	0.966	0.966	0.986	0.986	0.980	0.993	**1.000**	0.986	0.993	0.993
BN × BRp	0.966	0.616	0.664	0.623	0.616	0.815	0.548	0.493	0.664	0.541	0.569	0.980	0.980	0.959	0.986	0.966	0.966	0.973	**1.000**	0.993	0.993
BN × BF	0.959	0.610	0.658	0.616	0.610	0.808	0.541	0.514	0.658	0.548	0.562	0.959	0.959	0.966	0.966	0.959	0.973	0.966	0.966	**1.000**	1.000
BN × BT	0.959	0.610	0.685	0.630	0.623	0.822	0.527	0.500	0.685	0.548	0.562	0.973	0.973	0.980	0.980	0.973	0.986	0.993	0.980	0.973	**1.000**

## Data Availability

The data presented in this study are available on request from the corresponding author.

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
