# Peer review of "Comparison of Six Measures of Genetic Similarity of Interspecific Brassicaceae Hybrids F2 Generation and Their Parental Forms Estimated on the Basis of ISSR Markers"

_genes, 2024, doi:10.3390/genes15091114_

Round 1

Reviewer 1 Report

Comments and Suggestions for Authors  This paper explores the application of ISSR molecular marker technology to assess the genetic similarity between interspecies Brassicaceae F2 hybrids and their parental forms. The study employs a specific combination of materials that may not have been thoroughly reported in previous research, suggesting a degree of originality. However, the paper presents certain limitations regarding the depth and breadth of the research. Firstly, the study is confined to particular species and their hybrids, which may restrict the generalizability of the findings. Secondly, the research relies solely on specific molecular marker technology, omitting other potential technologies such as SNP markers, which could enhance the dimension and accuracy of genetic similarity analysis. Additionally, while the article addresses the applicability of various measurement methods, it does not provide an in-depth discussion of the applicability and limitations of each method.   Major comments:
(1)Check whether the number and selection of ISSR molecular markers used sufficiently represent the diversity of genotypes of the Brassicaceae family across generations, and evaluate their possible impact on the reliability of genetic similarity results?
(2)The selection of the six genetic similarity measures in the study seemed appropriate, taking into account the advantages of the different algorithms. However, the paper is not sufficiently detailed on the reasons for the choice of methods, comparisons between them, and their respective advantages and disadvantages.  
Especially when the results of the Rogers method are significantly different from those of other methods, the reasons and their impact on the research results should be explained in detail.
(3)The article points out that the two genetic similarity measurement methods, Sokal and Michener, and Euclidean, yield the same results. This phenomenon is briefly discussed.  The paper should provide more explanation and discussion of why the two results are consistent, and discuss whether this similarity reflects substantial biological information or is just a coincidence of computational methods. If the methods are similar in nature, it is necessary to discuss which method has more advantages in practical applications?
(4)UPGMA is a commonly used method in population genetic analysis, but it assumes that the evolutionary rate of all species is constant, which may not always be reasonable.  The authors are advised to elaborate on the rationale for choosing this method and discuss the impact that other clustering methods, such as neighborhood similarity or maximum likelihood methods, may have on the study results.  If possible, conducting a comparative analysis between these methods may give more confidence to the results.

Author Response

Response to Reviewer 1 Comments

Reviewer #1

Point 1: This paper explores the application of ISSR molecular marker technology to assess the genetic similarity between interspecies Brassicaceae F2 hybrids and their parental forms. The study employs a specific combination of materials that may not have been thoroughly reported in previous research, suggesting a degree of originality. However, the paper presents certain limitations regarding the depth and breadth of the research. Firstly, the study is confined to particular species and their hybrids, which may restrict the generalizability of the findings. Secondly, the research relies solely on specific molecular marker technology, omitting other potential technologies such as SNP markers, which could enhance the dimension and accuracy of genetic similarity analysis. Additionally, while the article addresses the applicability of various measurement methods, it does not provide an in-depth discussion of the applicability and limitations of each method.

Response: We would like to thank you very much for appreciating our research and the results from it. The evaluation of genetic similarity between interspecific hybrids of F2 Brassicaceae and their parental forms is very important to us, since we depend on the results obtained for further research on the material under consideration. Therefore, at this stage of the research we have limited ourselves to these specific genotypes. It seems to us that the results obtained can be generalized. We are aware that further research on other materials may expand knowledge and shed light on aspects that were only hinted at in our study. Our study is based solely on a specific molecular marker technology, but, as we discuss below, this does not detract from the significance of the results obtained.

Major comments:

Point 2: (1) Check whether the number and selection of ISSR molecular markers used sufficiently represent the diversity of genotypes of the Brassicaceae family across generations, and evaluate their possible impact on the reliability of genetic similarity results?

Response: Previous studies have shown that the type of markers did not affect the evaluation of genetic similarity coefficients of rapeseed genotypes. We added this information in the new version of the manuscript. The ISSR markers used sufficiently represent the diversity of genotypes of the Brassicaceae family and in this respect are a good tool for obtaining reliable results of genetic similarity and clustering of the genotypes studied. As for the number of markers, it is, of course, important in assessing genetic similarity. It must not be too small. "Small" is a relative term. The most important thing is that the markers should represent the entire genome as far as possible.

Point 3: (2) The selection of the six genetic similarity measures in the study seemed appropriate, taking into account the advantages of the different algorithms. However, the paper is not sufficiently detailed on the reasons for the choice of methods, comparisons between them, and their respective advantages and disadvantages.

Especially when the results of the Rogers method are significantly different from those of other methods, the reasons and their impact on the research results should be explained in detail.

Response: The choice of the six genetic similarity measures in our study was based on the fact that they are the most commonly used in practical research. Five of these methods (in addition to Rogers' method) are based on counting alleles present simultaneously in both genotypes being compared. The Rogers method takes into account the number of alleles present in only one of the compared genotypes, while the other methods are based on the number of alleles present simultaneously in both compared genotypes. This is explained in detail in the Discussion.

Point 4: (3) The article points out that the two genetic similarity measurement methods, Sokal and Michener, and Euclidean, yield the same results. This phenomenon is briefly discussed.  The paper should provide more explanation and discussion of why the two results are consistent, and discuss whether this similarity reflects substantial biological information or is just a coincidence of computational methods. If the methods are similar in nature, it is necessary to discuss which method has more advantages in practical applications?

Response: Two methods of measuring genetic similarity, Sokal and Michener and Euclidean, give the same results. We discussed this phenomenon only briefly out of respect for the scientists who introduced the formula, which went down in history with the names Sokal and Michener. The similarity, or in fact identity, with Euclid's method is due to the fact that the aforementioned authors 'adopted' Euclid's method by using a bivalent variable in it: zero and one.

Point 5: (4) UPGMA is a commonly used method in population genetic analysis, but it assumes that the evolutionary rate of all species is constant, which may not always be reasonable.  The authors are advised to elaborate on the rationale for choosing this method and discuss the impact that other clustering methods, such as neighborhood similarity or maximum likelihood methods, may have on the study results.  If possible, conducting a comparative analysis between these methods may give more confidence to the results.

Response: We have previously performed clustering not only with the UPGMA method, but also with other methods such as maximum likelihood, neighbor-joining, minimum-evolution and maximum parsimony. The results we obtained were very similar, and some were identical. In the manuscript, we chose to present only one clustering method, UPGMA. We have added this information in the revised version of the manuscript.

Reviewer 2 Report

Comments and Suggestions for Authors

Dear Authors, I have reviewed the manuscript and have the following comments:

The study presented compared six measures to calculate genetic similarity: Euclidian, Jaccard, KulczyÅ„ski, Sokal and Michener, Nei and Rogers. The research material included 21 oilseed rape genotypes (ten F2 generation interspecific Brassicaceae hybrids and 11 parental forms) and 146 alleles obtained with 21 ISSR molecular markers. 

The Abstract is not good, it is recommended to rewrite it with space for the material and methods, results and conclusions. 

The Introduction and Discussion are adequate. 

Results: only tables that are actually relevant to the manuscript should be left, what is not should be simplified or put in the Appendices. 

It is also worth highlighting the hypothesis, the results - and in this context it is also advisable to make the Conclusion chapter more concrete. 

Author Response

Response to Reviewer 2 Comments

Reviewer #2

Point 1: Dear Authors, I have reviewed the manuscript and have the following comments:

The study presented compared six measures to calculate genetic similarity: Euclidian, Jaccard, Kulczyński, Sokal and Michener, Nei and Rogers. The research material included 21 oilseed rape genotypes (ten F2 generation interspecific Brassicaceae hybrids and 11 parental forms) and 146 alleles obtained with 21 ISSR molecular markers.

Response: Thank you very much.

Point 2: The Abstract is not good, it is recommended to rewrite it with space for the material and methods, results and conclusions.

Response: Thank you very much for your attention to the Abstract. We have rewritten it and added more information about the results we received.

Point 3: The Introduction and Discussion are adequate.

Response: Thank you very much.

Point 4: Results: only tables that are actually relevant to the manuscript should be left, what is not should be simplified or put in the Appendices.

Response: We agree with the Reviewer that the results in the Tables are significant to the manuscript. We also believe that those shown in the Figures are also significant to the picture of the overall manuscript. Those shown in Figure One present, among other things, the ranges of similarity coefficient values and their scatter. Figure 2 shows the correlations between the values of similarity coefficients. And in Figure 3 we have presented the clustering of the genotypes studied. We believe that these Figures are important and make the manuscript complete. Therefore, we decided to leave them in the manuscript. If the esteemed Reviewer and Editor determine that these Figures should nevertheless be included in the supplementary materials, we will do so.

Point 5: It is also worth highlighting the hypothesis, the results - and in this context it is also advisable to make the Conclusion chapter more concrete.

Response: Thank you very much for your attention to the Summary of our research results. We have corrected the Conclusions section. The new revised version of this chapter is: “The purpose of this study was to compare six statistical measures (Euclidean, Jaccard, Kulczynski, Sokal and Michener, Nei and Rogers) for assessing the genetic similarity of 21 rapeseed objects (ten interspecific hybrids of Brassicaceae of generation F2 and 11 of their parental forms) based on observations of 146 alleles obtained using 21 ISSR molecular markers. All the mentioned measures define different categories of genetic similarity assessment and coexist with other techniques such as gene sequencing and flora and fauna analysis. Choosing the appropriate type of measure requires a thorough understanding of its purpose and calculation technique to provide an assessment that is precise and comprehensive enough. Significantly different results were obtained when using the Rogers measure. The research material was divided into two identical groups using any of the methods, despite differences in genetic similarity coefficient values. Two of the presented genetic similarity measures (Sokal and Michener, and Euclidean) were identical.”.

Reviewer 3 Report

Comments and Suggestions for Authors  

Congratulations on your article "Comparison of Six Measures of Genetic Similarity of Interspecific Brassicaceae Hybrids F2 Generation and Their Parental Forms Estimated on the Basis of ISSR Markers"!

Your study on genetic similarity is a significant contribution, with broad applications across various scientific disciplines.

In this research, you compared six methods for calculating genetic similarity: Euclidean, Jaccard, Kulczyński, Sokal and Michener, Nei, and Rogers.

The study analyzed 21 rapeseed genotypes (including ten interspecific Brassicaceae hybrids of the F2 generation and 11 of their parental forms) and 146 alleles, using ISSR molecular markers.

The results show that, despite differences in the values of genetic similarity coefficients, the research material was consistently divided into two identical groups using any of the proposed methods. Notably, the Sokal and Michener, and Euclidean measures produced identical results.

Suggestions:

1. Consider reducing the introduction to four relevant paragraphs.

2. In Tables 5-7, underline the diagonal (either by bolding the 1.00 values or adding a line).

3. For Figure 2, add the coefficient of multiple determination (R2) for each method: Euclidean, Jaccard, Kulczyński, Sokal and Michener, Nei, and Rogers.

Author Response

Response to Reviewer 3 Comments

Reviewer #3

Point 1: Congratulations on your article "Comparison of Six Measures of Genetic Similarity of Interspecific Brassicaceae Hybrids F2 Generation and Their Parental Forms Estimated on the Basis of ISSR Markers"!

Response: Thank you very much.

Point 2: Your study on genetic similarity is a significant contribution, with broad applications across various scientific disciplines.

Response: Thank you very much.

Point 3: In this research, you compared six methods for calculating genetic similarity: Euclidean, Jaccard, Kulczyński, Sokal and Michener, Nei, and Rogers.

The study analyzed 21 rapeseed genotypes (including ten interspecific Brassicaceae hybrids of the F2 generation and 11 of their parental forms) and 146 alleles, using ISSR molecular markers.

The results show that, despite differences in the values of genetic similarity coefficients, the research material was consistently divided into two identical groups using any of the proposed methods. Notably, the Sokal and Michener, and Euclidean measures produced identical results.

Response: Thank you very much.

Suggestions:

Point 4: 1. Consider reducing the introduction to four relevant paragraphs.

Response: Thank you very much for appreciating our manuscript. I must admit that it is very difficult for us to skip some paragraphs included in the Introduction. In writing them we felt that all the information we included was important and relevant to the topic of the research we undertook. It seems to us that to omit any passage would be to detract from the value of the manuscript. Therefore, let us leave the Introduction unchanged as to content. On the other hand, we have combined some paragraphs to form four, which form a coherent and (in our opinion) complete whole.

Point 5: 2. In Tables 5-7, underline the diagonal (either by bolding the 1.00 values or adding a line).

Response: In Tables 5-7, diagonals are highlighted by making the values 1.00 bold and changing the background color to gray.

Point 6: 3. For Figure 2, add the coefficient of multiple determination (R2) for each method: Euclidean, Jaccard, Kulczyński, Sokal and Michener, Nei, and Rogers.

Response: A multiple regression model was written for each method (Euclidean, Jaccard, Kulczynski, Sokal and Michener, Nei, and Rogers), where the independent variables were the remaining methods. For each model, the coefficient of multiple determination (R2) was calculated. Values of the coefficient of multiple determination (R2) for each method (Euclidean, Jaccard, Kulczynski, Sokal and Michener, Nei, and Rogers) are added in the caption of Figure 2.

Round 2

Reviewer 1 Report

Comments and Suggestions for Authors

The article has been revised as required

Comments on the Quality of English Language

There is no big problem in English writing

Reviewer 2 Report

Comments and Suggestions for Authors

Thank you, I rocommand it for publication.